# Effect of Red Light on the Expression of the Phytochrome Gene Family and the Accumulation of Glycoside Alkaloids in Potatoes

**DOI:** 10.3390/foods12234194

**Published:** 2023-11-21

**Authors:** Xiaolu Zhang, Hong Jiang, Weigang Liu, Ya Wang, Fankui Zeng

**Affiliations:** 1Research & Development Center for Eco-Material and Eco-Chemistry, Lanzhou Institute of Chemical Physics, Chinese Academy of Sciences (CAS), Lanzhou 730000, China; 18298441872@163.com (X.Z.); jianghong@licp.cas.cn (H.J.); liuweigang@licp.cas.cn (W.L.); 2College of Life Science and Engineering, Lanzhou University of Technology, Lanzhou 730050, China; wangya502@163.com; 3Yantai Zhongke Research Institute of Advanced Materials and Green Chemical Engineering, Yantai 264000, China; 4Qingdao Center of Resource Chemistry & New Materials, Qingdao 266000, China

**Keywords:** potato tubers, light irradiation, glycoside alkaloids accumulation, bioinformatics analysis, phytochrome gene

## Abstract

Potatoes are the fourth major food crop in the world. Higher levels of glycoside alkaloids (GAs) lead to detrimental effects on the edibility and processing qualities. GAs are largely influenced by light; however, the mechanisms of this regulation by light are not well understood. By analyzing the bioinformatics of the phytochrome genes (PHYs) in potatoes, its expression level, the content of GAs and the correlation between them under different lights, this study aims to reveal the specific mechanism of light-regulated GAs accumulation and provide a theoretical basis for improved potato processing. Results based on high-performance liquid chromatography and imaging mass microscopy showed that red light induced a significant increase in *α-*chaconine and *α-*solanine accumulation compared to white light, but there was almost no accumulation in the dark within 12 days. Meanwhile, a bioinformatic analysis of PHY gene family members was performed, and the results showed that the five *StPHYs* were distributed on chromosomes 1, 2, 5, 7 and 10, with amino acid counts ranging from 704 to 1130. *StPHYs* genes have abundant light-responsive elements. Also, the expression patterns of *StPHYs* were dramatically induced by red light. Additionally, a correlation analysis showed that the GAs accumulation was significantly correlated with *StPHYs* expression. This research is useful for comprehending the metabolism of GAs regulated by light and monitoring food safety in potatoes.

## 1. Introduction

The potato (*Solanum tuberosum* L.) is an annual dicotyledonous herb in the Solanaceae family that is native to South America. As the world’s fourth largest food crop, it is planted all over the world. It is rich in carbohydrates, essential amino acids, vitamins and minerals [1]. It also contains a large amount of dietary fiber that is beneficial for people with high blood lipids and hypertension [2,3]. However, there is a kind of secondary metabolite, known as glycoside alkaloids (GAs), which is prevalent in potato and has always been recognized as an anti-nutritional factor. GAs are found in the roots, stems, flowers and fruits of Solanaceae and Liliaceae [4], and were first detected in *Solanum nigrum* L. In the early 19th century, the isolation and identification of more than 100 GAs from plants were performed, and currently about 46 GAs, such as *α-*chaconine, α-solanine and *α*-tomatine, have been found in solanaceous species, for example, potatoes and tomatoes [5]. The chemical structure of GAs consists of hydrophilic oligosaccharide chains and hydrophobic glycosidic ligands. This special structure is able to inhibit the activity of acetylcholinesterase, which leads to the disruption of the function of cell membranes [6]. It is well known that GAs in potato tubers are highly affected by a series of environmental factors during storage, like light, temperature, humidity, wounding and other factors. Among them, light is a major factor that can induce GAs accumulation. The GAs content exceeding 20 mg/100 g can cause serious toxic reactions or even death, which has an adverse effect on fresh potatoes or potato-related products. Therefore, it is necessary to uncover the mechanism of GAs accumulation under light irradiation.

Light is perceived by various photoreceptors for the development of photosystems in plants. Four photoreceptors play a major role in the photomorphogenesis of potatoes: phytochrome (PHY), cryptochrome (CRY), phototropin (PHOT) and ultraviolet-B receptor (UVR) [7]. The phytochrome includes two states, including phytochrome FR-absorbing isomer (Pfr) and phytochrome R-absorbing form (Pr), and the inactive state (Pr) can convert to an activated state (Pfr) to function after receiving red light signals [8]. The Pfr-type phytochrome initiates photomorphogenesis by inhibiting phytochrome-interacting factors (PIFs) after light irradiation [9]. The PHY genes in *Arabidopsis thaliana* contain PHYA, PHYB, PHYC, PHYD and PHYE subfamilies, respectively. PHYA receives continuous far-red light at wavelengths of 700–750 nm, and is unstable in response to light, while PHYB, PHYC, PHYD and PHYE receive red light from 600 to 700 nm and are highly photostable [10]. Upon activation by red light, photosensitive pigments are transferred from the cytoplasm to the nucleus, where they trigger photomorphogenesis responses by inhibiting PIFs [11].

PHY A has been reported to participate in regulating the carbon flux through a series of metabolic pathways, like glycolysis and the tricarboxylic acid (TCA) cycle [12]. Also, Kozuka et al. [13] found that phytochrome plays a key role in plant de-yellowing by coordinating the degradation of oil body lipids and sugar production, which are involved in the process of cotyledon greening during de-yellowing. GAs is synthesized from the mevalonate (MVA) pathway, the major pathway of cholesterol synthesis, which belong to the precursors of SGAs. It has been reported that this pathway is regulated by phytochromes in *Arabidopsis* [14]. In *Arabidopsis*, both phytochrome genes have been found to contribute to chloroplast formation in R and FR light [15]. Under light exposure, a GA level of 3.0–17.1 mg/100 g fresh weight (FW) could be detected in potato tubers, which exceeds the newly introduced limit of 10 mg/100 g according to safety guides. Altogether, light has been found to play a decisive role in the accumulation of GAs in potato tubers [16]. Although light-induced GA accumulation in potato tubers is well established, the regulation mechanism of GA induction upon exposure to light, especially red light, remains to be elucidated.

## 2. Materials and Methods

### 2.1. Preparation of Plant Materials

Potato cv. Atlantic, Favorita, EV and Gan Yin No. 3 were grown in the Potato Research Institute in Dingxi city, Gansu Province, China. These cultivars were cultivated in an experimental fileds that are uniformly managed. Organ samples and tubers were collected and harvested in August and September in 2023, respectively. After that, the obtained samples were taken to the laboratory and stored at 4 °C for experimental analysis.

### 2.2. Light Irritation Treatment and Sampling

One hundred and fifty potato tubers were washed and air dried at room temperature (22 ± 2 °C). Then, these tubers were divided into three groups randomly. For red-light treatment, the potato tubers were placed in a light-tight box and provided with constant light using light-emitting diodes (LEDs) for up to 12 days. The wavelength of red light was 660 nm with ± 10 nm bandwidth; the other group was exposed to white light (400–680 nm), and tubers placed in darkness were used as control. Samples from the tuber epidermis of about 3–4 mm depth and the organs of roots, stems, leaves and flowers were immediately frozen in liquid nitrogen and then stored at −80 °C for glycosidic alkaloid extraction and total RNA extraction. The samples with three replications were taken at 0 d, 3 d, 6 d, 9 d and 12 d, and stored for further use.

### 2.3. Extraction and Detection of GAs

Glycoside alkaloids were extracted and detected according to the method of Zeng et al. (2017) with some modifications [17]. The frozen potato samples were ground in 5% acetic acid at the ratio of 1:40, sonicated for 30 min and filtered at room temperature. Then, the filtrate was resuspended by adding 40 mL of 5% acetic acid, and the filtrates were combined. All the filtrate was combined with 10 mL of concentrated ammonia to adjust the pH to 10. The obtained solution was placed in a 70 °C water bath for 50 min, and then kept at 4 °C overnight. After that, the solution was centrifuged at 18,000 r/min for 10 min at 6 °C, and the precipitate was washed twice with 2% (*v*/*v*, 1:10) ammonia. Then the precipitate was dried under vacuum conditions, dissolved in a tetrahydrofuran/acetonitrile/20 mmol KH_2_PO_4_ solution with a ratio of 1:3 and filtered through a microporous membrane for HPLC analysis. HPLC separations were performed with a Shimadzu HPLC system coupled with a chromatographic column of Inertsil NH_2_ (5 μm, 4.0 mm × 250 mm) using a mobile phase consisting of acetonitrile/KH_2_PO_4_ (80:20, *v*/*v*) at a flow rate of 1 mL/min. Standards of *α-*chaconine of 95% (ChromaDex, CDXP-19-00326, Los Angeles, CA, USA) and α-solanine of 95% (ChromaDex, CDXP-20-00076) were used for the LC analyses.

### 2.4. GA Observation via an Imaging Mass Microscope

The imaging MS analysis of GA accumulation was based on the method of Deng et al. (2021) with some modification [18]. Epidermal part of potato tuber that had been treated for 0, 3, and 12 days were selected, quickly frozen in liquid nitrogen. The selected portions were cut into 20 μm sections at −18 °C using a cryomicrotome (Leica CM1950, Nussloch, Germany). And then thawed on the electrically conductive glass slides for matrix coating and imaging. The selected matrix was 2,5-dihydroxybenzoic acid (DHB) and the method of matrix coating was airbrushing. After coating the matrix solution (50 mg/mL), which dissolved in MeOH and distilled water at a ratio of 7:3, on the prepared sections, an image mass microscopy system (Shimadzu, Tokyo, Japan) equipped with an optical microscope, an atmospheric pressure chamber source and a hybrid quadrupole ion trap time-of-flight analyzer was used to collect the MS imaging data. Visualization and relative quantification were performed by the imaging MS Solution^®^ Version 1.30 software (Shimadzu, Tokyo, Japan).

### 2.5. Identification and Analysis of StPHY Genes

Identification of the *StPHY* genes was carried out as described previously by Zhang et al. (2020) [19]. Based on the whole genome database of potatoes, we searched the typical protein structural domains of the PHY gene by the accession numbers PHY (PF00360), PAS (PF00989) and GAF (PF01590) to obtain sequence fragments of the *StPHY* genes, and then bioinformatically analyzed them through the online website according to the amino acid sequences of PHYs. MEGA X was used to construct the phylogenetic tree; the TBtools v1.09832 tool was used to draw the chromosome distribution map of the gene and analyze the *StPHY* genes structure, the promoter cis-acting elements of *StPHYs* and the promoter cis-acting elements.

### 2.6. The Expression Pattern of StPHYs Genes

The relative expression of the *StPHYs* were analyzed using a 2 × SYBR Green qPCR Mix kit (No. AH0104-B, Sparkjade, Jinan, China). RNA was extracted from the roots, stems, leaves, flowers and tubers of different potato varieties (Favorita, Atlantic, EV, Gan Yin 3), and then the RNA was reverse-transcribed into single-stranded cDNA using a reverse transcription kit (No. AG0305-B, Sparkjade, Jinan, China). Primers for the PHY family genes were designed using Primer Premier 5 software, and the sequences of the primers are shown in detail in Appendix A. The potato EF-1α gene was used as the internal reference gene for expression analysis. Quantitative Real-time PCR(qRT-PCR) analysis was performed through a Light Cycler 96 SW 1.1 instrument according to the following conditions: 94 °C for 180 s, with 1 cycle; 95 °C for 30 s, 60 °C for 20 s, 72 °C for 30 s with 45 cycles, and finally an extension step for 30 s at 72 °C. The relative expression of the *StPHYs* were calculated using the 2^−ΔΔCt^ method [20].

### 2.7. Statistical Analysis

All the data are presented as the means ± SE, which were also analyzed for statistical significance by Ducan’s multiple range test using SPSS 20.0 software at *p* < 0.05. A correlation analysis was carried out in Origin 2021 software.

## 3. Results

### 3.1. GA Analysis by Mass Spectrometry Imaging (MSI)

We observed variations in the spatial distribution and relative content changes of four glycoside alkaloids in the potato periderm. The distributions of four kinds of GAs, including dehydrochaconine, *α-*chaconine, dehydrosolanine and α-solanine under different light treatments can be clearly viewed in the optical images of potato periderm that are shown in Figure 1. GAs were almost undetectable on day 0 (Figure 1a). This indicates that under the storage conditions of keeping in dark, the fresh potato periderm contains few GAs. After 3 d of light treatment, sporadic fluorescent dots appeared on the mass spectrometry imaging map (Figure 1b,c). With increases in storage time, they showed clear distribution characteristics in potato tubers (Figure 1d,e). In terms of MSI color scans, the brighter the color, the higher the GA content in the region. The GA distribution was the most obvious after 12 d of red-light treatment. According to the published study of Dhalsamant et al. [21], the distribution of GAs in potatoes was not homogeneous; the concentrations usually decreased from the periderm to the center of the tuber, and most GAs are located within the 1 mm of the outer surface of the tuber. In line with the previously published findings, we also saw that the periderm of potato tubers generated a significant amount of GAs following red-light treatment [22].

### 3.2. GA Detection in Potato Tubers via HPLC

The regulation of GA content by light is particularly obvious in contrast with other environmental factors, and photoreceptors can accurately sense light signals in the environment and regulate their synthesis. Phytochrome, as the red photoreceptor, plays a major role in potato photomorphogenesis [7]. Therefore, we investigated the effectiveness of red light (RL) and white light (WL) compared to the dark (DK) in inducing GA synthesis in ‘Atlantic’ tubers. The results showed that GA contents gradually increased with the light irritation time. Red light induced the highest level of GAs at any time during storage, followed by white light. However, a few GAs were detected in tubers that kept in the dark for 12 days (Figure 2), which is consistent with the results of MSI (Figure 1). GA concentrations were 2–5-fold higher after red light exposure than in darkness after 12 d of light treatment, which is in agreement with the results of Okamoto’s study [23].

### 3.3. Identification of Potato PHYs

A search of the potato genome-wide database (Solanum tuberosum v6.1) for PHY genes yielded five members of the PHY gene family, namely *StPHYA*, *StPHYB*, *StPHYB2*, *StPHYC* and *StPHYE*, whose relevant physicochemical properties are shown in detail in Table 1. *StPHYs* gene family members encoded proteins with amino acid numbers ranging from 704 to 1130 and molecular weights ranging from 77,442.38 to 125,782.06 Da, among which the *StPHYE* protein had the lowest amino acid number and the smallest molecular weight. The CDS lengths of *StPHYs* genome members range from 1467 to 2040 bp, and the isoelectric point (PI) sizes range from 5.68 to 5.8, indicating that *StPHY* proteins are acidic proteins. The total average hydrophobicity index ranges from −1.071 to −0.111, indicating a low hydrophilicity. Subcellular localization showed that the *StPHYs* genes are all nuclear genes (Table 1).

### 3.4. Analysis of StPHYs Gene Structure

UTR and CDS of varying lengths and numbers were observed between *StPHYs* family members, all of which are break genes and separated by introns. The *StPHYs* family possesses six conserved domains: PAS_2 (PF08446), PAS (PF00989), HATPase_c (PF02518), GAF (PF01590), PHY (PF00360) and HisKA (PF00512). All members contain the typical PAS-GAF-PHY domain at the N-terminus and three structural domains: PAS_2, HisKA and HATPase_c (Figure 3). This result means that PHY genes with similar motif distributions had similar gene structures.

### 3.5. Phylogenetic Analysis of StPHYs

In order to understand the affinities and evolutionary distances between *StPHY* genes, *StPHY* amino acid sequences were download and subjected to the following BLAST (by NCBI) search with other PHY species. A total of 36 PHY proteins were obtained from 10 species, including *Solanum tuberosum*, *Arabidopsis thaliana*, *Pisum sativum*, *Nicotiana tabacum*, *Sorghum bicolor*, *Zea mays*, *Oryza sativa*, *Solanum lycopersicum*, *Vitis vinifera* and *Capsicum annuum.* The phylogenetic evolutionary tree was constructed and is shown in Figure 4. The five PHY homologs were clustered into four clades: PHYA, PHYB, PHYC and PHYE. All five *StPHY* proteins were closely related to those of Solanaceae. *StPHYB* is more closely related to *SlPHYB1*, *SlPHYB2* and *NtPHY*B, whereas *StPHYA* is closer to *SlPHYA* and *CaPHYA*; *StPHY* is closer to *SlPHYE*; and *StPHYC* is closer to *SlPHYC* and *VvPHYC* (Figure 4).

### 3.6. Promoter Cis-Acting Element Analysis of StPHYs

To further explore the potential functions of *StPHY* family genes, cis-acting elements of the PHY gene promoter region (2000 bp upstream) were predicted in Tbtools v.1.09876 software. A total of 16 elements were screened and identified, and could be categorized as light-responsive cis-elements, phytohormone-responsive cis-elements, environmental-stress-responsive cis-elements and transcription factor binding sites. Hormone response elements include the auxin element, the salicylic acid response element, the gibberellin response element and the abscisic acid response element. Light-responsive cis-elements were most abundant in the environmental stress response. The results revealed that light-responsive elements had the largest proportion in the five potato *StPHY* family genes. The ABA-responsive cis-elements were the most numerous among the phytohormone-like responses (Figure 5). These results indicate that potato *StPHY* gene expression is regulated primarily by light signaling and plays an important role in photomorphogenesis.

### 3.7. Expression Analysis of StPHYs Genes in Different Organs from Different Potato Varieties

PHY gene expression varies with different varieties and the expression of the individual gene also varies with different organs. All members of the *StPHYs* family expressed the most in flowers, and in the “EV” cultivar, which displayed significant expression levels except for *StPHYE.* Instead, all expressions in the stems and leaves were the lowest in contrast with roots and flowers. Additionally, *StPHYA* and *StPHYE* in stems and leaves showed a higher level in the “Favorita” variety, while *StPHYB* levels in stems and leaves of the “EV” variety was significantly higher than the others. Therefore, the expression of these *StPHY* genes is tissue-specific and variety-specific (Figure 6).

### 3.8. Expression Analysis of StPHYs Genes under Different Lights

According to the induction of GA accumulation by light, the expression of photoreceptors PHYs in potato tubers were further analyzed. The results showed that the relative expression levels of all five genes were up-regulated under white and red-light treatment. When the tubers were exposed to white light, all PHY gene expressions were increased after 3 d of irradiation, except for *StPHYA* and *StPHYB*, which peaked after 9 d of white-light treatment. Among them, *StPHYC* showed the most significant response to white light. As for red-light treatment, all *StPHYs* members displayed an obvious increment throughout the period in contrast with darkness. *StPHYA* and *StPHYB2* peaked after 6 d when exposed to red light, whereas *StPHYB/C/E* increased gradually along with the light exposure time, which reached the maximum level after 12 d of red-light treatment (Figure 7).

### 3.9. Correlation Analysis between StPHYs Expression and GA Contents

The correlation between PHY gene expression and *α-*chaconine and α-solanine contents under red-light treatment was assessed using the Sperman correlation coefficient. *StPHYB* gene expression was highly significantly correlated with the content of two kinds of GAs (*p* < 0.01); the correlation coefficients were 0.74 and 0.75, respectively. In addition, *StPHYC* was also significantly correlated with α-chaconine and α-solanine contents, with a correlation coefficient of 0.68 and 0.7. However, *StPHYA* and StPHYB2 expressions were negatively correlated with GA content (*p* < 0.05). It is hypothesized that the red-light receptor protein *StPHYs* is involved in GA accumulation in potato tubers and it may be a major regulator (Figure 8).

## 4. Discussion

Light is an important environmental factor in the regulation of potato growth and the content of GAs [24]. The duration of light exposure significantly impacted the increase in the content of GAs in potato tubers, and their content increased in a parabolic manner [25]. In addition, the accumulation of GAs in potato tubers varied with light quality; when the light intensity was less than 500 μ mol·m^2^ s^−1^, the accumulation of GAs increased with increasing light intensity, whereas if it was more than 750 μ mol∙m^2^ s^−1^, it decreased with increasing light intensity [26]. Interestingly, light exposure induces genome demethylation and the accumulation of solanine in the tubers, suggesting that the level of DNA methylation is positively correlated with the solanine content [27]. The results of an HPLC assay in our study showed that both light treatments enhanced the accumulation of GAs, and the fastest accumulation was observed in red-light-treated tubers, followed by white light. In order to accurately and visually observe the changes in GA contents, we further performed MSI analyses. We found that the number of fluorescent dots in potato tubers treated with red light at 12 days similarly increased dramatically. In addition, the content of α-chaconine was more than that of α-solanine at different times, indicating that *α*-chaconine is a prevalent GA, as described by Friedman [28]. In the study of Mekapogu et al., ‘Atlantic’ cultivars responded differently when exposed to different light wavelengths, including purple, red, blue, green, yellow and UV light. Our results exhibited a similar phenomenon and revealed that red light plays a decisive role in the accumulation of GAs in ‘Atlantic’ tubers [29].

Phytochromes are important in light-induced development throughout plant growth, including flowering induction, shade avoidance response and seed germination. As a protein superfamily in plants, the family members and copy numbers vary in different plants. The PHYs of model plant *Arabidopsis* belong to the PHYA, PHYB, PHYC, PHYD and PHYE subfamilies, respectively, with the PHYB and PHYD subfamily genes showing a higher degree of similarity, which is explained by the fact that the PHYD subfamily was derived from the PHYB subfamily in the last whole-genome duplication event in *Arabidopsis* [30]. There are only three phytochromes, including PHYA, PHYB and PHYC, in *Oryza sativa* and *Zea mays*, while there are five phytochrome in potatoes, including PHYA, PHYB, PHYB2, PHYC and PHYE, which are localized on five different chromosomes (Appendix A). The chromosomal localization had a symmetrical distribution, which indicated that each clade may have divergent or redundant gene copies. Our findings indicate that gene duplication has an important role in the expansion and evolution of potato *StPHYs*. With the exception of the *StPHYC* gene, all of the genes had high gene densities at both ends of the chromosome. All the phytochromes share a photosensory core module, consisting of PAS, GAF and PHY domains and a bilin chromophore [31].

The potato PHY gene family encompasses two physical types: photolabile type I and photostable type II. Concomitantly, according to the gene structure and sequence similarity, *StPHY* can be further subdivided into four subgroups homologous to PHYA/B/C/E progenitors. In the present study, we found that the structural domains in the *StPHYs* proteins are highly similar in type and arrangement. The distribution and functions of the structural domains of PHYs are consistent with the previous results, which can serve as a platform for protein interactions and a model for responding to changes in light conditions, oxygen levels and redox potentials [32]. The N-terminal photoreceptor domain covalently binds chromophores, and is related by the photochemical properties of phytochrome, while the C-terminal photoregulatory domain is mainly involved in the formation of phytochrome dimers and downstream signaling processes [33]. We identified 10 conserved motifs in the *StPHYs* gene, which were identical in number and type and aligned in the same way, showing a strong similarity. In this study, we also found differences in the positional distribution of the six structural domains of all members, which may account for the differences in the expression patterns of the members of the PHY gene family (Appendix A).

A total of 16 cis-regulatory elements were identified within the putative promoter regions of genes encoding the potato PHYs gene family; they are responsive to phytohormones, abiotic stresses and photoresponses. PHY genes, as important light signaling receptor proteins in plants, play important roles in plant growth and development, morphogenesis, secondary metabolism and stress tolerance [34]. PHYA is a type of photo-unstable protein that breaks down rapidly under light and is involved in the early red-light response, whereas PHYB, PHYC, PHYD and PHYE are photostable proteins that play a major role under sustained red or white light [35]. In our study, it was found that the potato’s GA content peaked after 12 days of red-light treatment, which may be the result of the induced PHYs expression.

In our work, the transcript abundance of *StPHYA* and *StPHYB* varied in different organs, and the expression levels of individual genes displayed differences among different cultivars, these results showed that the expression patterns of *StPHY* genes were complex (Figure 6). After light exposure, all *StPHY* members were induced by white light and red light; however, the increases induced by red light were the most significant (Figure 7), showing that red light is dominant in regulating GA accumulation. Importantly, *StPHYB/C/E* were found to have the most obvious response to red light irradiation. According to the correlation analysis, the correlation coefficients between *StPHYB/C* and the α-solanine content were 0.75 and 0.68, while for α-chaconine they were 0.75 and 0.7. Thus, the GA content was significantly and positively correlated with the *StPHYs* gene expression. This result also indicated that the *StPHYs* gene play a more critical role in regulating GA biosynthesis. However, the detailed regulatory mechanism of *StPHYs* genes in response to light signaling and GA accumulation needs to be further investigated in future studies.

## 5. Conclusions

In this study, five *StPHY* genes were identified and their biological functions were clarified. Only the *StPHYD* subfamily member is missing from the *StPHY* gene family in potatoes. Although the functional structures are similar, there are obvious differences in expression patterns. Red light affects the content of GAs, and the photoreceptor protein *StPHYs* are involved in the regulation of GA content in potatoes. The availability of the *StPHYs* gene facilitates the exploration of the protein function of the gene and the mechanism of action of GA accumulation in response to different light signals. In addition, our research further refined the molecular regulatory mechanism of GAs. The study of photoregulation mechanisms in GA biosynthesis holds promise for targeted interventions of signaling pathways to reduce or even block them, providing a solution to the problem of greening and wasting of potatoes due to the accumulation of GAs.

## Figures and Tables

**Figure 1 foods-12-04194-f001:**
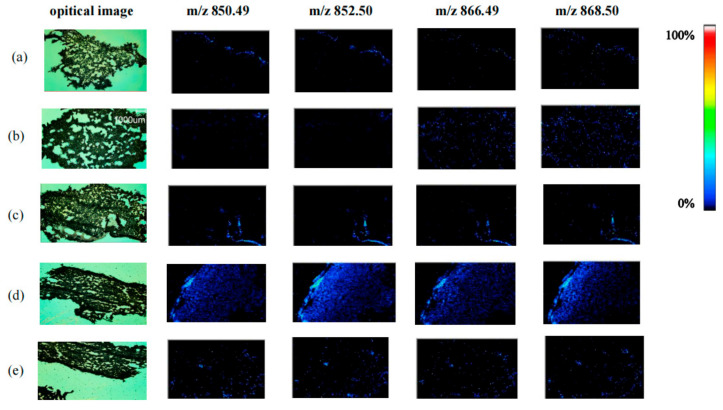
The optical images of potato tubers and mass spectrometric images of four GAs: dehydrochaconine ([M + H]+ at *m*/*z* 850.49), α-chaconine ([M + H]+ at *m*/*z* 852.50), dehydrosolanine ([M + H]+ at *m*/*z* 866.49) and α-solanine ([M + H]+ at *m*/*z* 868.50) under red-light treatment at different times ((**a**) 0 day, (**b**) 3 day, (**d**) 12 day) and dark treatment ((**c**) 3 day, (**e**) 12 day).

**Figure 2 foods-12-04194-f002:**
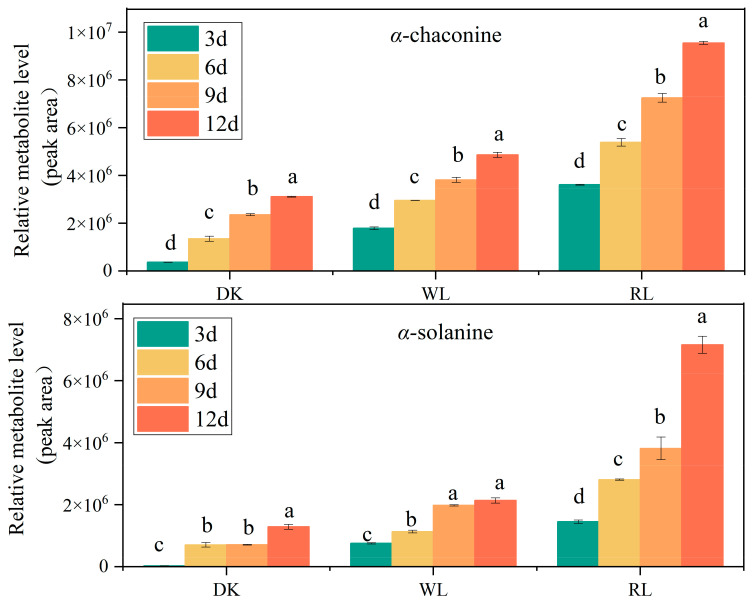
Effect of light treatment on α-chaconine and α-solanine accumulation in potato tubers. Data shown are means ± SD (*n* = 3 independent biological replicates). Different letters indicate significant differences between different times according to Duncan’s multiple tests.

**Figure 3 foods-12-04194-f003:**
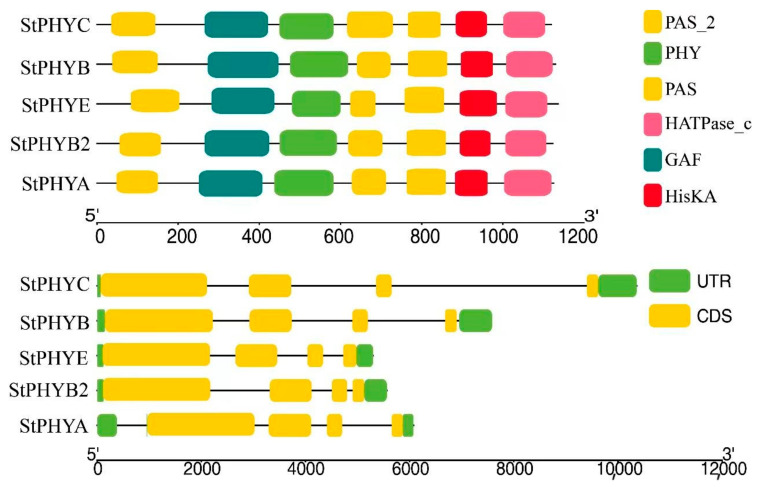
The conserved motifs and gene structures of *StPHYs*.

**Figure 4 foods-12-04194-f004:**
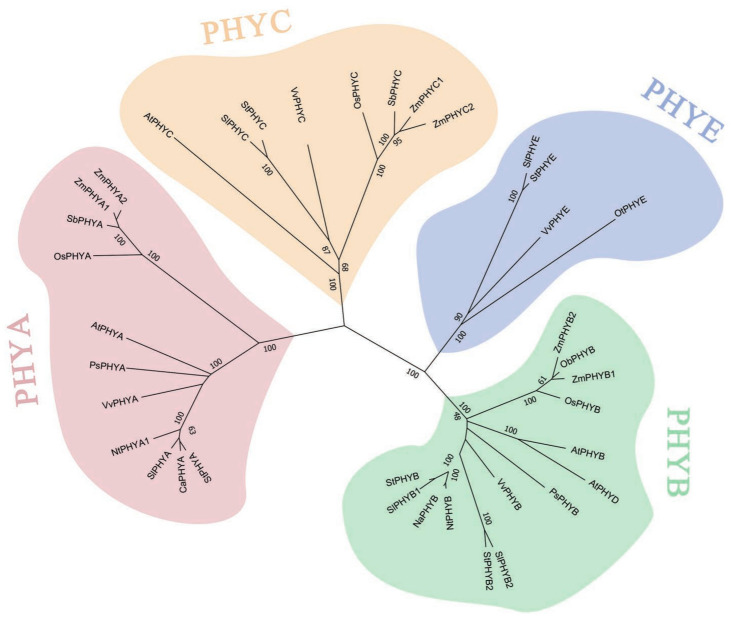
Phylogenetic relationship of PHY proteins in different plant species. (The numbers in the phylogenetic tree represent bootstrap values, which are used to test the credibility of the branch, and the values range from 0–100).

**Figure 5 foods-12-04194-f005:**
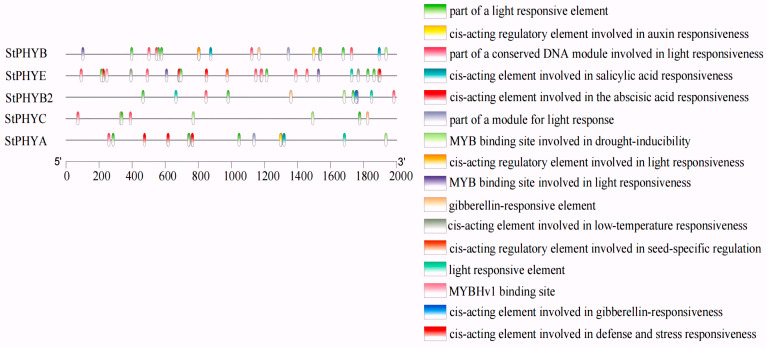
Cis-elements in the promoters of PHY genes that are related to stress responses.

**Figure 6 foods-12-04194-f006:**
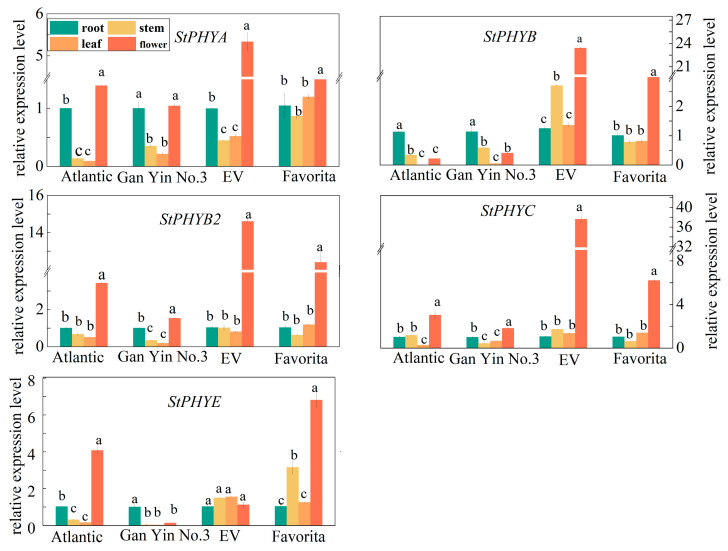
Relative expression levels of *StPHYs* in different organs of four species (Atlantic, GanYin 3. Favorite and EV). (Values are given as means ± standard error from three biological replicates. Different letters indicate significant differences between different varieties according to Duncan’s multiple tests).

**Figure 7 foods-12-04194-f007:**
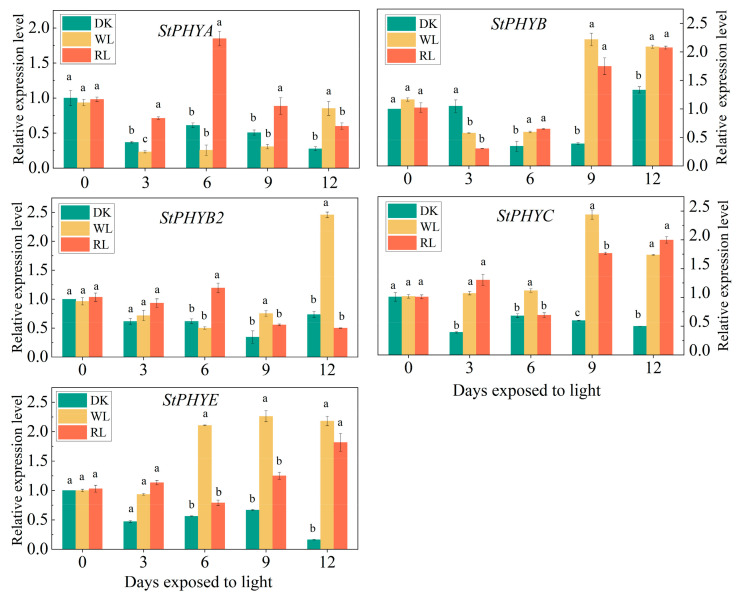
Analysis of relative expression levels of five *StPHYs* gene family members in potato peels. (DK: dark process, WL: natural light process, RL: red light process (650 nm). Values were normalized against the expression data of *StPHYs* and are given as means ± standard error from three biological replicates. Note: Different lowercase letters on the bars indicate significant differences in the expression level between different treatments).

**Figure 8 foods-12-04194-f008:**
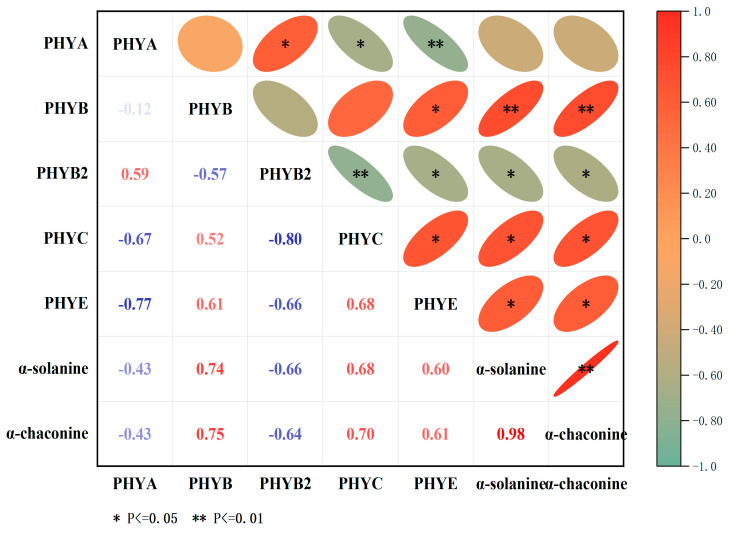
Pearson’s correlation coefficient between the relative expression levels of *StPHYs* and GAs contents (*α*-chaconine, *α*-solanine) under red-light treatment. (The degree of correlation is shown by color tones, with red numbers denoting positive correlation and blue numbers denoting negative correlation.) ** represents an extremely significant correlation (*p* < 0.01). * represents a significant correlation (*p* < 0.05).

**Table 1 foods-12-04194-t001:** Characteristics of phytochrome (PHY) genes.

Sequence ID	Number of Amino Acid	Molecular Weight	pI	Instability Index	GRAVY	Subcellular Localization
PHYC	1120	124,731.52	5.89	47.76	−0.111	Nucleus
PHYB	1130	125,364.58	5.68	43.98	−0.141	Nucleus
PHYE	704	77,442.38	5.84	53.61	−0.171	Nucleus
PHYB2	1125	125,782.06	5.75	46.74	−0.145	Nucleus
PHYA	1123	124,671.98	5.87	42.05	−0.147	Nucleus

## Data Availability

Data are contained within the article or Appendix A.

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
