# Peer review of "Effect of Red Light on the Expression of the Phytochrome Gene Family and the Accumulation of Glycoside Alkaloids in Potatoes"

_foods, 2023, doi:10.3390/foods12234194_

Round 1
Reviewer 1 Report
Comments and Suggestions for Authors
The present manuscript describes the study conducted to elucidate the specific mechanism of light-regulated accumulation of glycoside alkaloids, which may help improve potatoes' food safety. It is a very interesting article and fits perfectly in the journal's scope. However, after reading the document carefully, I think that the authors may address the following issues to strengthen the manuscript:
1. I think it could be interesting to include in the introduction and discussion some of the next articles
· Agriculture 2020, 10(5), 139; https://doi.org/10.3390/agriculture10050139
· Plant Physiology and Biochemistry 2022, 170, 218-224; https://doi.org/10.1016/j.plaphy.2021.12.007
2. Line 99. How long was the sample sonicated?
3. Line 102. What was the final pH of the sample?
4. Line 103-104. Was the sample centrifuged at room temperature?
5. Line 104. What was the ratio between the weight of the precipitate and the volume of 2% ammonia used?
6. Line 105-106. What was the ratio between the weight of the precipitate and the volume of tetrahydrofuran/acetonitrile/20 mmol KH2PO4 solution used?
7. Figure 7 and Figure 8. are too small and cannot be read clearly.
Author Response
Dear Editors and Reviewers,
We would like to thank you and the reviewers for the helpful comments and suggestions on our manuscript (foods-2666623) entitled "Effect of red light on the expression of PHYS gene familys and accumulation of Glycoside alkaloids in potato". The comments are valuable for revising and improving our paper. We have carefully revised the manuscript according to the comments and have done our best to respond to the suggestions. A point-by-point response is provided as below.
Response to Reviewer#1’s comments:
The present manuscript describes the study conducted to elucidate the specific mechanism of light-regulated accumulation of glycoside alkaloids, which may help improve potatoes' food safety. It is a very interesting article and fits perfectly in the journal's scope. However, after reading the document carefully, I think that the authors may address the following issues to strengthen the manuscript:
- I think it could be interesting to include in the introduction and discussion some of the next articles.
Response: We will use these articles in line 298–303 for our discussion, as suggested.
The content was changed to "Light is an important environmental factor in the regulation of potato growth and the content of GAs [26]. The time of light exposure significantly impacted the increase in the content of GAs in potato tubers, and its content increased in a parabolic manne. The accumulation of GAs in potato tubers varied with light quality, and at light intensities of L ≤ 500 μmol·m2 s-1, the accumulation of GAs increased with increasing light intensity, whereas at L ≥ 750 μmol·m2 s-1, the concentration of GAs decreased with increasing light instead [27]. Besides, light exposure induces genome demethylation and the accumulation of solanine in the tubers, suggesting that the level of DNA methylation is positively correlated with the solanine content [28]. "
2.Line 99. How long was the sample sonicated?
Response: The sample in experiment required 30 minutes to sonicate. According to the suggestion, this content is added in line 104 of the article. According to the suggestion, this content was added in line 104 of the article.
3.Line 102. What was the final pH of the sample?
Response: The final pH of sample is 10. According to the suggestion, this content was added in line 106 of the article.
4.Line 103-104. Was the sample centrifuged at room temperature?
Response: The sample was centrifuged at 6°C. According to the suggestion, this content was added in line 108 of the article.
5.Line 104. What was the ratio between the weight of the precipitate and the volume of 2% ammonia used?
Response: The weight of the precipitate and have a volume ratio of 1:10.
6.Line 105-106. What was the ratio between the weight of the precipitate and the volume of tetrahydrofuran/acetonitrile/20 mmol KH2PO4 solution used?
Response: The ratio between the weight of the precipitate and the volume of tetrahydrofuran/acetonitrile/20 mmol KH2PO4 solution is 1:3.
- Figure 7 andFigure 8 are too small and cannot be read clearly.
Response: We have modified the charts to make them as clear as possible.

Reviewer 2 Report
Comments and Suggestions for Authors
What was the reason for choosing these specific potato varieties (Atlantic, Favorita, EV, and Gan Yin No. 3) for the study? Were there any characteristics or traits of interest associated with these varieties?
Could you provide more details about the uniform management practices applied to these cultivars during their growth? Were there any specific cultivation techniques or conditions that were standardized across all cultivars?
Could you please provide insights into why a 12-day duration was specifically chosen for the red-light treatment of the potato tubers? Was there a particular biological or experimental reason for selecting this time frame, and how does it relate to the study's objectives?
Could you please provide more details on the specific light intensity thresholds where GAs accumulation was observed to increase or decrease, and what is the significance of these thresholds in the context of your study's findings?
Could you please provide further insights into the specific roles of the different phytochrome gene family members in potatoes? How do these gene family members influence light signaling and the synthesis of glycoside alkaloids (GAs) in potatoes?
The text mentions the need for further investigation into the regulatory mechanisms of the StPHYB gene in light signaling and GAs synthesis. Please elaborate on potential future research directions in this regard.
Author Response
Dear Editors and Reviewers,
We would like to thank you and the reviewers for the helpful comments and suggestions on our manuscript (foods-2666623) entitled "Effect of red light on the expression of PHYS gene familys and accumulation of Glycoside alkaloids in potato". The comments are valuable for revising and improving our paper. We have carefully revised the manuscript according to the comments and have done our best to respond to the suggestions. A point-by-point response is provided as below.
- What was the reason for choosing these specific potato varieties (Atlantic, Favorita, EV, and Gan Yin No. 3) for the study? Were there any characteristics or traits of interest associated with these varieties?
Response: For these varieties, they are special varieties planted by the Potato Research Institute in Dingxi, Gansu Province. Among them, Atlantic and Favorita are prevailing and dominant varieties in the world. And another two varieties were new-cultivated varieties in Northwest China, which are mainly used to process starch or dehydrated powder.
- Could you provide more details about the uniform management practices applied to these cultivars during their growth? Were there any specific cultivation techniques or conditions that were standardized across all cultivars?
Response: They are grown in the plant areas that are uniformly managed, e.g., the same rainfall of 400mm-600mm and cultivated by fertilizing 135kg pure nitrogen per hectare, 75 kg pure phosphorus per hectare, and 105 kg pure potassium per hectare during growing.
- Could you please provide insights into why a 12-day duration was specifically chosen for the red-light treatment of the potato tubers? Was there a particular biological or experimental reason for selecting this time frame, and how does it relate to the study's objectives?
Response: The duration of light exposure had a major effect on the potato tubers' increased GA content. The macroscopic phenotypic outcomes of several light treatments for potatoes are depicted in followed figure. After three days, it was noted that the potato's skin treated with red light began to become green. Despite the fact that glycoside alkaloid synthesis and chlorophyll synthesis were two different mechanisms, there was a positive correlation between them. So, the time form 0 d to 12 d was selected to investigate the potential mechanism. So we sampled the epidermis every three days after it turned green.
- Could you please provide more details on the specific light intensity thresholds where GAs accumulation was observed to increase or decrease, and what is the significance of these thresholds in the context of your study's findings?
Response: According to earlier research, the amount of GAs accumulated in potato tubers varied depending on the quality of the light. At light intensities of L ≤ 500 μ mol·m2 s-1 , the amount of GAs accumulated increased with increasing light intensity, but at light intensities of L ≥ 750 μ mol·m2 s-1 , the concentration of GAs decreased with increasing light. However, the detailed action of specific light intensity thresholds where GAs accumulation under red light is still unknown, and we will focus on this field in future work.
- Could you please provide further insights into the specific roles of the different phytochrome gene family members in potatoes? How do these gene family members influence light signaling and the synthesis of glycoside alkaloids (GAs) in potatoes?
Response: Researches from the past has demonstrated that phytochrome gene of potato has diverse functions, for example, StPHYA is crucial for seed germination, de-yellowing, and inhibition of hypocotyl elongation under far-red light. Additionally, StPHYA is involved in photoperiod sensing in light-grown potato plants and in resetting the circadian clock that controls leaf movements. StPHYB also inhibits hypocotyl elongation under red light, and mutations of PHYB can cause precocious flowering, significant petiole lengthening, and stunted leaf development. StPHYC plays a less significant role in red light, but in conjunction with StPHYB, StPHYD, and StPHYE, it modulates the shade avoidance response and controls plant growth under light.
As for these gene family members affecting light signaling and the synthesis of glycoside alkaloids (GAs) in potatoes, the detailed mechanism can be conjectured to explain from two perspectives. One is the phytochrome-interaction factors, such as PIF, HY5, which can affect the Glycoside alkaloids accumulation by binding to the promoter of target genes of glycoaloid metabolism (GAME1, GAME4, GAME17), affecting the steady-state concentrations of transcripts coding for SGA pathway enzymes (Wang et al., 2018). Another is phytohormones, such as ABA, Auxin, JA and GA and so on, these phytohormones can influence the interplay between steroid and triterpenoid biosynthetic pathways, thereby affecting the production of steroidal alkaloids (Kuo et al., 2012; Markowski et al., 2022). Based on the present study, we will further revealed the regulation mechanism step by step in combination with existing reports.
- The text mentions the need for further investigation into the regulatory mechanisms of the StPHYB gene in light signaling and GAs synthesis. Please elaborate on potential future research directions in this regard.
Response: There is increasing concern about glycosidic alkaloids' safety. Our research revealed that the glycosidic alkaloid content of potatoes was positively correlated with the expression levels of PHYs, The obtained information could serve for breeding project of low glycosidic alkaloid varieties with specific quality which comply with consumer demand. Additionally, the detailed and clear mechanism of light-regulation GAs accumulation needs to be revealed by future researches.

Reviewer 3 Report
Comments and Suggestions for Authors
The manuscript is scientifically valuable and may have practical applications in terms of finding directions for solutions to global problems in ecological management in terms of wasting potatoes as a result of the accumulation of greenhouse gases.
Using modern techniques (high performance liquid chromatography and mass spectrometry imaging), the authors examined the molecular regulatory mechanism of glycoside alkaloids (GA) undesirable in the human body (anti-nutrients), depending on storage conditions, focusing on different light ranges. Compared to white light, it was shown that red light-induced a significant increase in GA accumulation in potatoes stored for 12 days. Bioinformatics analysis of PHY gene family members was also performed.
The manuscript has a good structure, I have no comments regarding the content. The presented results were subjected to statistical analysis and discussed using relevant literature from the years 1989-2022, including nearly 30% from the last years 2020-2022.
However, attention should be paid to the editing quality of the manuscript. There are missing or excess spaces in many places, and some figures have too small a font, reducing their readability. In References, the method of describing literature sources rather differs from the journal's requirements, because the journals' titles are bold instead of the articles' publication year. In the methodology and figure captions, it is "P < 0.05", and in the text of the manuscript "p < 0.05". The bracket in the figures caption is rather unnecessary. It is worth checking the correctness of "L ≤ μ 500 mol·m2s-1" and "L ≥ 750·μmol·m2s-1" (lines 296-297).
Which figure is "Figure A2" referring to? (line 334).
Comments on the Quality of English LanguageThe quality of the English Language is good, only requires minor corrections.
Author Response
Dear Editors and Reviewers,
We would like to thank you and the reviewers for the helpful comments and suggestions on our manuscript (foods-2666623) entitled "Effect of red light on the expression of PHYS gene familys and accumulation of Glycoside alkaloids in potato". The comments are valuable for revising and improving our paper. We have carefully revised the manuscript according to the comments and have done our best to respond to the suggestions. A point-by-point response is provided as below.
Response to Reviewer#3’s comments:
There are missing or excess spaces in many places, and some figures have too small a font, reducing their readability. In References, the method of describing literature sources rather differs from the journal's requirements, because the journals' titles are bold instead of the articles' publication year. In the methodology and figure captions, it is "P < 0.05", and in the text of the manuscript "p < 0.05". The bracket in the figures caption is rather unnecessary. It is worth checking the correctness of "L ≤ 500 μ mol·m2s-1" and "L ≥ 750·μmol·m2s-1" (lines 296-297). Which figure is "Figure A2" referring to?
Response:
(1)We have expanded the too-small numbers in the article and fixed the space problem .
- The format of the references has been changed according to the requirements of the journal,
- Figure A2 is incorrectly written in Figure S2, a supplement to the manuscript we submitted in line 325 and 340.Line 340 of Figure A2, a supplement to the text we submitted, We have changed to "Figure S2 ".
(4)To be consistent with the article case, the correlation coefficient p labelled in Figure 9 has been changed to uppercase and modified as follows:
(5)Following a thorough investigation and adjustment, the optical density flux unit in lines 296-29 was modified to "L ≤ 500 μ mol·m2 s-1 " and "L ≥ 750 μ mol·m2 s-1"
